# Current Trends and Characteristics of Hepatocellular Carcinoma in Patients with Autoimmune Liver Diseases

**DOI:** 10.3390/cancers13051023

**Published:** 2021-03-01

**Authors:** Eirini I. Rigopoulou, George N. Dalekos

**Affiliations:** Department of Medicine and Research Laboratory of Internal Medicine, National Expertise Center of Greece in Autoimmune Liver Diseases, General University Hospital of Larissa, 41110 Larissa, Greece; eirigopoulou@med.uth.gr

**Keywords:** autoimmune liver diseases, hepatocellular carcinoma, autoimmune hepatitis, hepatic, primary biliary cholangitis, primary sclerosing cholangitis

## Abstract

**Simple Summary:**

While hepatocellular carcinoma (HCC) is inextricably linked to underlying cirrhosis due to chronic viral hepatitis, alcohol consumption, and non-alcoholic fatty liver disease, the role of autoimmune liver diseases (AILDs) as risk factor has been disputed. In this review we present data showing AILDs to confer increased risk for HCC development, albeit lower than other hepatic diseases. We also highlight several risk factors that indicate patients with autoimmune hepatitis and primary biliary cholangitis at high and moderate risk for HCC development, including cirrhosis, older age, male sex, and co-existing factors, such as alcohol consumption and treatment response. As surveillance is of utmost importance for early diagnosis and subsequent effective treatment, we suggest refinement of screening strategies according to prevailing risk factors in patients with AILDs.

**Abstract:**

Hepatocellular carcinoma (HCC), the commonest among liver cancers, is one of the leading causes of mortality among malignancies worldwide. Several reports demonstrate autoimmune liver diseases (AILDs), including autoimmune hepatitis (AIH), primary biliary cholangitis (PBC), and primary sclerosing cholangitis (PSC) to confer increased risk of hepatobiliary malignancies, albeit at lower frequencies compared to other liver diseases. Several parameters have been recognized as risk factors for HCC development in AIH and PBC, including demographics such as older age and male sex, clinical features, the most decisive being cirrhosis and other co-existing factors, such as alcohol consumption. Moreover, biochemical activity and treatment response have been increasingly recognized as prognostic factors for HCC development in AIH and PBC. As available treatment modalities are effective only when HCC diagnosis is established early, surveillance has been proven essential for HCC prognosis. Considering that the risk for HCC is not uniform between and within disease groups, refinement of screening strategies according to prevailing demographic, clinical, and molecular risk factors is mandated in AILDs patients, as personalized HCC risk prediction will offer significant advantage in patients at high and/or medium risk. Furthermore, future investigations should draw attention to whether modification of immunosuppression could benefit AIH patients after HCC diagnosis.

## 1. Introduction

Hepatocellular carcinoma (HCC) is the commonest malignancy among primary liver cancers and according to data from 2018 is sixth and fourth in rank in incident and death cases, respectively, among cancers worldwide [1,2,3]. In detail, about 841,000 new HCC cases and 782,000 deaths were expected to have occurred in 2018 worldwide [1].

A wide geographical variability in HCC incidence has been long recognized, which mainly depends on prevalent risk factors in areas of high endemicity [4]. The most characteristic example is that of hepatitis B virus (HBV) infection that has accounted for the most HCC cases in Asian and Sub-Saharan populations, while aflatoxin exposure is documented to amplify this risk in areas of increased exposure [5].

HCC is inextricably linked to the presence of chronic liver disease, with up to 90% of cases having liver cirrhosis [3,4]. Accordingly, a recent meta-analysis showed a marked increase in HCC incidence for a variety of hepatopathies, including hepatitis C virus (HCV), HBV, primary biliary cholangitis (PBC), autoimmune hepatitis (AIH), and non-alcoholic fatty liver disease (NAFLD), when the disease progresses from the non-cirrhotic to cirrhotic stage [5,6]. Furthermore, in 40% of cirrhotic patients, development of HCC is the cause of death [3,4].

Data derived from a systematic review demonstrated 56% and 10% of HCC cases to be attributed to HBV and HCV infection, respectively [7]. HBV-related HCC has been shown to aggregate in less developed countries and specifically in Eastern Asia and Sub-Saharan Africa, while HCV-related HCC aggregates in more developed countries, including North America and North Africa [7]. Still, the epidemiology of HCC is evolving, and, accordingly, significant changes have been reported during the preceding years attributed to alterations in prominent risk factors for HCC. An in-depth analysis of data on the global burden of liver cancer highlighted the significant increase in global liver cancer incidence and number of liver cancer cases during a period spanning from 1990 to 2017 [8].

Consequently, HCC mortality has also been increasing. As shown in a study from the US, mortality related to HCC increased between 2000 and 2016 in both sexes, though the burden of HCC in males was double of that attributed to females. Moreover, increasing trends were observed in adults aged 65–74 and those over 75 [9]. 

Moreover, HCC cases were reported to have a decreasing trend in high incident areas and an increasing trend in low incident areas [4]. Fundamental for the reported decrease in HCC incidence has been the implementation of HBV vaccination in children in the developing countries. This was initially demonstrated in Taiwan and emphasized subsequently in other countries, including China, Alaska, and Thailand [10]. Still, a population-based study from Taiwan conducted 30 years after the implementation of HBV vaccination has shown that HBV incidence declined significantly in children and young adults, though increased rapidly after the age of 30 and peaked for both sexes in the seventh decade of life [11]. Several studies have also shown effective antiviral treatment to have contributed toward reducing the risk of HCC in patients with chronic HBV infection [12,13,14,15].

On the contrary, HCV is nowadays considered the key contributor of HCC increase at a global level [8]. Of interest, the incidence of HCC in HCV-related liver cirrhosis is much higher than that in AIH-related cirrhosis. Several studies have elucidated up to the present several direct and indirect interactions of HCV with cellular components in cirrhotic liver that promote hepatocarcinogenesis [16]. Furthermore, data in experimental animal models have shown HCV proteins, including HCV core, NS3, NS4B, and NS5B, to have oncogenic properties and eventually transform specific cells types [17,18]. The new direct-acting oral antivirals have considerably improved survival of patients with HCV infection, though the risk of hepatocarcinogenesis does nοt seem to have been eliminated entirely [19].

Alcoholic cirrhosis is another well-established risk factor for HCC [20]. Moreover, as obesity and insulin resistance are rising especially in the Western world, the burden of NAFLD and its complications, including HCC, is increasingly recognized and is expected to increase further in the near future [21,22]. Even more, both alcohol and obesity are important co-founding factors for HCC in patients with other hepatopathies. Based on changes on various risk factors and trends recorded, the number of new HCC cases and deaths is estimated to rise significantly until 2040 [23].

Although epidemiological trends and clinical characteristics of HCC related to viral hepatitis, alcohol, and NAFLD are well acknowledged, such data in autoimmune liver diseases (AILDs) are scarce and often conflicting.

## 2. Autoimmune Liver Diseases

AILDs, including AIH, PBC, primary sclerosing cholangitis (PSC), and their variant forms, encompass a wide spectrum of liver diseases with different pathophysiology, clinical presentation, treatment options, and prognosis. They can co-occur either in a synchronous or metachronous fashion [24,25], while recognition of coexisting chronic liver diseases, including viral hepatitides, alcohol, NAFLD, and drug-induced liver disease is increasingly documented [26,27,28,29]. 

AIH and PBC are classic autoimmune diseases characterized by progressive hepatocyte and cholangiocyte damage, respectively, leading to cirrhosis especially in those diagnosed late during the disease course or left untreated [30,31,32,33,34,35]. PSC is an immune-mediated cholestatic disease with complex features, including strong association with inflammatory bowel disease [36,37]. 

Growing evidence suggests autoimmune diseases to confer increased risk of hepatobiliary malignancies, albeit at lower frequencies in AIH and PBC patients compared to other liver diseases. A recent population-based, case-control study from the US evaluating in- and outpatient adults older than 65 years has shown autoimmune diseases to be associated with hepatobiliary malignancies [38]. The most striking association in this study was that of HCC for PBC patients (odds ratio (OR): 31.33 (95%CI: 23.63–41.56)) and intrahepatic cholangiocarcinoma (CCA) (7.53 (5.73–10.57)), extrahepatic CCA (5.59 (4.03–7.75)), gallbladder cancer (2.06 (1.27–3.33)), and ampulla of Vater cancer (6.29 (4.29–9.22)) for PSC patients [38]. Another population study from New Zealand demonstrated patients with AILDs to have decreased survival compared to the general population. Amongst them, AIH and PSC patients had significantly increased risk for hepatic and extra-hepatic malignancies, respectively [39].

Until recently, there were discrepant views related to the need for HCC surveillance in patients with AILDs. This relates to the fact that AILDs are relatively rare compared to other previously reported, well-established causes of HCC, and existing data are often inadequate to exert reliable conclusions based on small sample size, short follow up, lack of control populations, and absence of reported incidence of HCC in several studies of patients with AILDs [40]. 

The role of surveillance is to establish HCC diagnosis at an early stage in order to ensure increased survival [41]. Indeed, curative treatments, including liver resection (LR), radiofrequency ablation (RFA), and liver transplantation (LT) are restricted to cases with early HCC [3,42,43].

Extensive discussions in the past have dealt with the threshold of HCC incidence accepted as significant to initiate surveillance in patients with chronic liver disease. Scientific societies have embraced the proposal, based on cost-effectiveness, suggesting surveillance to be applied in populations where yearly HCC incidence is higher than 1.5–3.0% [44]. We need to appraise this threshold with skepticism, as data to support cost-effectiveness have been deduced from viral hepatitis patients and have not been validated in other cirrhotic populations. This is of immense importance as different underlying hepatopathies and other interrelated factors designate variations in the HCC risk they confer.

Accumulating data has led both the European Association for the Study of the Liver (EASL) and the American Association for the Study of Liver Diseases (AASLD) in their latest clinical practice guidelines to advise screening mainly in cirrhotic patients with AIH and PBC [43,45]. No other parameters have been recognized as substantial to be incorporated in proposed guidelines. As AILDs are considered rare diseases, a systematic approach of validated factors conferring risk for HCC in these patients is enforced.

The aim of this review is to present data related to epidemiology and characteristics of HCC in patients with AILDs, alongside prognostic features and treatment modalities.

### 2.1. Autoimmune Hepatitis

AIH is a heterogeneous disease, the main features of which are the presence of circulating autoantibodies, hypergammaglobulinemia, and interface hepatitis on liver histology [30,31,32]. Accumulating evidence suggests the incidence of AIH to be significantly rising, as shown in a population-based study from England during the period 1997–2015 [46] and studies from other countries, mainly from Northern Europe [47,48,49,50].

The perception of AIH being a rare disease affecting mainly middle-aged women has been challenged during the last decade, based on the increasingly reported number of cases of both sexes, all age groups, and all races [30,31,32,51,52]. The spectrum of the clinical manifestations of the disease spans from acute presentation to chronic liver disease, often co-existing with other autoimmune diseases (especially with Hashimoto’s thyroiditis) [30,31,32,51,52,53]. At the time of diagnosis, around one-third of patients with AIH already have cirrhosis [30,31,32,54,55,56]. Considering the heterogeneity of the disease’s features and the absence of disease-specific characteristics, AIH diagnosis is often challenging [28,29,30,31,32,57]. Implementation of diagnostic scoring systems has merely facilitated the diagnostic process by identifying under-recognized or under-diagnosed cases [58,59].

Several studies have reported impaired survival of AIH patients, which mostly relates to the presence of cirrhosis at diagnosis or later during follow up or ongoing inflammatory activity during follow up, leading to liver-related death, including development of HCC [39,49,60,61,62].

Early studies from the 1970s have shown immunosuppressive treatment to result in improved survival [63,64]. The prognosis of patients with active AIH is closely related to timely initiation of immunosuppressive treatment, while a succeeding remission is also essential [30,31,32]. 

Population-based studies have provided evidence proving the high mortality of AIH patients. As reported in a Danish nationwide cohort study of 2904 AIH patients, representing the whole AIH population, the 20-year risk of HCC was increased compared to controls, though the absolute risk was 1.6% [65].

A recent study from England spanning 18 years of follow up in 1008 AIH patients reported the 10-year cumulative all-cause mortality and the liver-related mortality, including HCC, to be 31.9% and 10.5%, respectively. The 10-year mortality risk from HCC in this study was reported to be 1.6% (95%CI: 0.8–3.1%) [46].

Even though AIH has been linked to the development of HCC, the lower risk (<3%) of HCC in these patients compared to those with other hepatopathies has also been confirmed by several other studies (Table 1) [47,49,50,55,62,66,67,68,69,70,71,72,73,74,75,76]. 

Data on the evolving trend related to the incidence of HCC in AIH patients are inconsistent, as attested by a population-based study from Denmark demonstrating doubling of HCC incidence during the study period [47], while the incidence was reported to decline in a retrospective study from the US [67]. This considerable variation in HCC incidence among AIH patients is highlighted in a meta-analysis of 25 studies that demonstrated moderate heterogeneity. This study reported a pooled HCC incidence of 3.06 (95%CI: 2.22–4.23) per 1000 person-years, with a range spanning from 0 to 11.34 per 1000 person-years [77]. This wide span rather relates to diverse characteristics of the source populations and study design. 

Several reports have analyzed individual factors that were associated with the development of HCC in AIH (Table 2). Similar to other chronic liver diseases, the most decisive feature for HCC development has been the presence of cirrhosis. A recent meta-analysis encompassing a wide spectrum of liver diseases reported cirrhosis to increase the risk for HCC by 2.79 times (from 0.19% to 0.53%) in AIH patients [6]. Accordingly, the incidence of HCC was lower in non-cirrhotic AIH patients (1.14 per 1000 patient-years, 95%CI: 0.60–2.17) compared to those with cirrhosis (10.07 per 1000 patient-years, 95%CI: 6.90–14.70) [77].

A prospective study from King’s College London has shown HCC to have developed in 6.2% of the whole AIH population and 12.3% of cirrhotic patients, with a rate of 1.1% per year [62]. Hence, the incidence of HCC is reportedly rising in patients being cirrhotic for more than five years [75,80]. The implication of cirrhosis as a risk factor for HCC has been highlighted in numerous other studies so far related to patients both from the West and the East [6,65,67,70,72,74,76].

Among other factors, older age has been another feature that has been associated with the development of HCC in AIH patients [67,76,77]. Several reports have specified male sex to be a risk factor for HCC, though this is not a unifying finding in all studies [47,49,62,70,77]. Moreover, a study from three European centers did not prove race to be a predisposing factor [81].

Increased aminotransferase levels—either in a continuous fashion or at last follow up, being a reliable indicator of ongoing necroinflammatory activity in non-responders to immunosuppressive treatment—have been indicated as a risk factor for the HCC in several studies [68,72]. Besides, more than two relapses during immunosuppressive treatment have been also identified as prognostic factors for HCC development [73]. Lack of disease remission during treatment was also indicated as one of the risk factors for HCC in a recent retrospective study of 355 AIH patients [79]. These findings clearly highlight sustained remission as a result of immunosuppression in AIH patients to be mandatory for long-term prognosis. Accordingly, Choi et al. have shown that patients with early (<1 year) combined normalization of aminotransferases and IgG exert the lowest rate of liver-related adverse outcomes, including HCC [76].

Immunosuppressive treatment for more than three years has been proposed as a risk factor of HCC in a retrospective study of AIH [70]. Defective immune-mediated tumor surveillance control as a result of chronic immunosuppression has been ascribed as a risk factor for various malignancies, especially in patients after organ transplantation. However, this is not supported by data from the majority of published studies on AIH. On the contrary and as previously stated, long-standing cirrhosis, alongside inadequate treatment response, are proven vital risk factors for HCC in AIH patients. Therefore, the current guidelines support surveillance in AIH patients with cirrhosis [30,31,32].

Up to the present, there is no information available on whether differences between type 1 and type 2 AIH patients in the incidence of HCC exist. In addition, no data exist on the incidence of HCC development in the rare group of HCV-induced AIH, which has been mainly documented in the previous era of interferon-alpha treatment [82].

Theoretically, co-occurrence of AIH with other chronic liver disease can synergistically increase the incidence of HCC in these patients. For instance, concomitant alcohol consumption was found to associate with a 6-times increased risk of HCC in AIH patients [78], though this study was published only as an abstract, while recently, another study did not find any difference [29]. NAFLD is nowadays the commonest liver disease worldwide, and its burden on the overall incidence of advanced liver disease and its complications, including HCC, is expected to increase exponentially during the forthcoming years [21,83]. NAFLD can also co-occur with AIH, but solid data on the long-term outcome of these patients are lacking. Indeed, so far, there was only a large retrospective Japanese study including 1151 AIH patients, with almost 17% of them with concurrent NAFLD, which showed that HCC and extrahepatic malignancies did not differ between patients with AIH alone and those with the AIH/NAFLD variant [84]. In parallel with the abovementioned findings, the first results from a recent large multicenter study from the International AIH Group in 583 patients (21.6% with concurrent NAFLD) showed no differences on progression to cirrhosis and liver related deaths, including HCC or LT [85]. In contrast, a small retrospective study from the US in 73 consecutive AIH patients showed that a subgroup of these patients had higher risk of experiencing adverse clinical outcomes with decreased survival, but this conclusion was based only on 22 patients with AIH/NAFLD [86]. These studies did not provide any information on clinical characteristics of patients with AIH/NAFLD that developed HCC during follow up.

### 2.2. Primary Biliary Cholangitis

PBC is a cholestatic liver disease of autoimmune etiology affecting mainly women, its main characteristics being the presence of antimitochondrial antibodies (AMA) and progressive destruction of small intrahepatic bile ducts with the potential to evolve to progressive fibrosis and finally to hepatic failure [33,34,35,87,88].

Compared to its initial description, PBC incidence and prevalence rates are reported to be on the rise [33,34,35,89,90,91,92]. The disease is most prevalent in North Europe and North America, where it was originally studied, with estimated incidence and prevalence rates ranging from 3.3 to 32 per million person-years and 19 to 402 per million, respectively [89,90,91,92], though lower prevalence and incidence rates have been recorded in South Korea [93]. High prevalence rates have been also described during the last decade in South Europe, a characteristic example being that of Central Greece [34]. Moreover, a systemic review from the Asian-Pacific region anticipated the overall prevalence to be higher than expected (118.75 cases per million (95%CI: 49.96–187.55)) with significant differences across regions [94].

Earlier diagnosis, due to increased disease awareness in conjunction with advances in PBC diagnosis, and the wide use of ursodeoxycholic acid (UDCA) have changed the natural history of PBC immensely during the preceding decades [95]. A considerable body of evidence over the years has shown that the use of UDCA (13–15 mg/Kg/day) is associated with survival advantage when administered in patients at early disease stages (stage I/II) who demonstrate biochemical response [96,97]. While early UDCA treatment leads to survival analogous to the general population [97,98], UDCA non-responders have the worst prognosis [99]. A recent multicenter study of 4805 PBC patients spanning 44 years of follow up has shown PBC to be diagnosed at an older age, though at earlier biochemical and histological stages, during recent decades [95].

This study clearly illustrated for the first time the increased rates of UDCA response as well as lower decompensation rates and higher 10-year transplant-free survival rates in PBC patients [95]. Based on these data, we could effortlessly extrapolate that the incidence of HCC has also been declining during these decades. Nevertheless, published results on the incidence of HCC in PBC patients vary considerably. A systematic review of 17 studies with high heterogeneity published between 1984 and 2011 demonstrated significantly higher HCC risk for the 16,368 PBC patients included, compared to the general population pooled relative risk (RR): 18.80 (95%CI: 10.81–26.79) [100], whereas no increased relative risk was found for other type of cancers. 

Following this systemic analysis, several large population-based studies have reported on the incidence of HCC in PBC. In a large Japanese study based on two national surveys involving 2946 patients, the incidence of HCC was 2.4% [101]. In a population-based cohort of 992 PBC patients from the Netherlands a 9-fold increased risk of developing hepatobiliary malignancy (standardized incidence ratio (SIR): 9.4; 95%CI: 3.04–21.8) over a 36 year follow up period was recorded [102]. In a low prevalence PBC area such as South Korea, the incidence of HCC in a PBC population of 2824 patients was 1.4% [93]. Trivedi and colleagues reported an incidence rate of 3.4 cases/1000 patient-years in a multicenter study of 4565 PBC patients [103]. A more recent systematic review and meta-analysis of 29 studies including 22,615 PBC patients reported the incident rate of HCC to be 4.17 per 1000 patient-years (95%CI: 3.17–5.47) [104].

Several factors have been linked to an increased risk for HCC in PBC patients (Table 3). In this last meta-analysis, cirrhosis at baseline was a strong risk factor for HCC, as attested by an incidence of 15.7 per 1000 patient-years (95%CI: 8.73–28.24) in those with cirrhosis compared to an incidence of 2.68 per 1000 person-years in non-cirrhotics [104].

The decisive role of cirrhosis in the development of HCC was highlighted in a comparison study from Spain, where stage III and IV PBC patients had comparable incidence of HCC to cirrhotic HCV patients (11.1% vs. 15%) [105]. Comparable data were reported in a prospective study from Japan, a cooperative study from Spain and Italy, where advanced-stage PBC (stage III/IV) was associated with development of HCC, and a cohort study from Greece [106,108,110].

Even though female predominance is strong in all PBC studies, males with PBC have shown higher frequency of HCC in most studies [101,107,108,109]. This male predilection for HCC in PBC is not supported by all reports [34,93,102]. There have been various assumptions on why such a predilection for HCC exists in male PBC patients. Estrogen-mediated inhibition of cytokines, including interleukin-6 exerting a protective role against hepatocarcinogenesis in females, has been proposed as a possible mechanism [113,114]. Furthermore, PBC is often overlooked in males, as the disease is considered a female disease, and an increase in γGT is often ascribed to other possible causes (i.e., alcohol consumption). Another reason for females getting earlier diagnosed with PBC is that females present more frequent than males with PBC-related symptoms, including pruritus and fatigue [115,116]. Accordingly, these are conceivable causes often leading to consideration of the disease at older ages in males. Indeed, existing data show men to get the diagnosis of PBC at older ages and with more advanced disease stage [34,116].

According to existing data, past HBV infection [107] and increased alcohol consumption in males may contribute toward higher incidence of HCC in male PBC patients [34,112]. Other predisposing factors for HCC were co-existing diabetes mellitus and body mass index (BMI) ≥ 25 (kg/m^2^) in two studies on Chinese PBC patients [107,112]. These studies from East Asia showcase the need for evaluation of individual risk factors for HCC according to the population under investigation. Of note, so far, BMI and concurrent diabetes mellitus have not been identified as risk factors for HCC development in patients with AIH. On the other hand, obesity as well as alcohol have also been reported as risk factors for the development of HCC in other chronic hepatopathies, including HBV and HCV infections [117]. Based on these facts, differences in risk factors for HCC between AIH and PBC patients may probably indicate differences in populations studied and subsequently in differences in individual characteristics of patients included in the studies, rather than discrepancies related to specific hepatopathies per se.

It has been widely demonstrated during the last decade that biochemical response to treatment is associated with improved prognosis, as highlighted by improved transplant-free/overall survival [33,118,119]. However, whether treatment response has any effect on the risk of developing HCC in PBC patients has been addressed in few studies and data provided are conflicting.

In a meta-analysis, Natarajan and colleagues reported similar HCC risk between patients on UDCA treatment and those without therapy, though stratification according to presence or absence of cirrhosis could not be done [104]. The most important findings originate from an international collaborative study analyzing data from 4565 PBC patients, demonstrating the most significant factor for HCC development being non-response to 12-month UDCA course (evaluated by Paris I criteria) irrespective of gender and disease stage [103]. Similar findings have already been reported in a previous smaller study (*n* = 375) from the Netherlands [111]. Currently, both EASL and AASLD recommend surveillance for HCC in PBC patients with cirrhosis [33,45].

We believe, though, that all data that we were previously referring to clearly highlight that there is space for an update on recommendations on surveillance strategies in PBC patients. Further risk stratification of PBC patients according to gender, co-existing hepatopathies, and biochemical response to UDCA treatment is mandated. Another important factor that needs to be implemented into surveillance tools is the quick transition between biochemical disease stages during the follow up period. This was recently suggested by a multicenter study in which half of early stage PBC patients progressed to more severe biochemical stages in five years and that this was associated with an increased risk of adverse clinical events [120].

### 2.3. Primary Sclerosing Cholangitis

PSC is an immune-mediated chronic cholestatic liver disease affecting intra- and/or extrahepatic bile ducts, often progressing to end-stage liver disease [121]. The disease is characterized by high heterogeneity in terms of clinical presentation, including large duct PSC, representing the classic form of the disease, small duct PSC, IgG4-related PSC, and the PSC/AIH variant [36]. PSC is often associated with inflammatory bowel disease. Contrary to classic autoimmune diseases, PSC does not have a pathognomonic autoantibody signature and does not respond to immunosuppressive treatment [121]. Despite previous assumptions of male predominance in PSC patients, recent large multicenter studies of the International PSC Study Group in more than 7000 PSC patients showed that the disease occurs as commonly in females as in males, albeit a more subclinical and favorable course is reported in females [122,123]. Currently, liver transplantation represents the only effective treatment tool.

The incidence of PSC varies greatly, following a North to South gradient. Studies from the preceding two decades have pointed toward an increase in PSC incidence [124,125].

A large body of evidence has identified PSC as a risk factor for hepatobiliary malignancies, CCA accounting for the majority of these cases [126]. The presence of PSC increases the risk for CCA by 160 times compared to the general population [126]. The cumulative incidence of CCA during the natural history of PSC is reported between 10% and 15%. Up to 50% of CCA cases are diagnosed in the first year after diagnosis, while the incidence rate remains constant thereafter at 1.5% per year [125]. Patients with PSC/inflammatory bowel disease (IBD) are reportedly being at even greater risk for development of CCA during the follow up [125,127,128].

On the contrary, most of the studies consider the risk of HCC in PSC rather negligible. In a study of 604 patients representing almost the entire PSC population of Sweden, HCC was identified in none of the patients during the follow up and in 4% of liver explants (4 out of 92 explanted livers) [126]. In a large study incorporating 502 PSC patients with a total of 4202 patient-years follow up, none developed HCC amongst the 119 cirrhotic patients [129]. Moreover, no HCC was identified in explanted livers from 140 patients who underwent liver transplantation [129]. Accordingly, in two studies from the Netherlands encompassing 590 and 211 patients, respectively, no HCC cases were reported during the follow up or in the explanted livers [125]. In contrast, a recent study from the UK involving IBD patients developing PSC during a 10-year follow up, found an HCC frequency of 1.8% (47/2588) among PSC/IBD patients [130]. In addition, a high incidence of HCC cases in PSC was reported in a study of 830 patients from the US. In fact, this study showed the highest incidence of HCC, as 20 patients (2.5%) were diagnosed with HCC during a median follow up of 9.5 years [131]. All HCC patients had cirrhosis. The higher incidence of HCC could be attributed to the longer follow up of this study. Even though the incidence of HCC is considered low in PSC patients, these 2 studies stress the need for implementing surveillance not only for CCA but also for HCC in PSC patients.

### 2.4. Variant Syndromes

Data on the prevalence of HCC in patients with variant syndromes are scarce, considering that detailed information on this subgroup of patients is missing in the majority of studies. In a study from 31 centers in the Netherlands, AIH patients had similar incidence of HCC (7/1110) compared to AIH/PBC patients (1/113), while none of 80 AIH/PSC patients developed HCC [50].

## 3. Screening for HCC in AILDs Patients

According to current guidelines, screening should be performed in all cirrhotic Child–Pugh A and B stage patients and Child–Pugh C stage patients awaiting liver transplantation independent of etiology [43,45,132]. There are no screening guidelines up to the present specific to any of the AILDs. Hence, abdominal ultrasound is recommended as the first line surveillance tool for HCC screening, while computed tomography (CT) and magnetic resonance (MR) imaging have been used as complementary tools to establish HCC diagnosis in cases where abdominal ultrasound has demonstrated low sensitivity, such as early HCC cases, obesity, and alcohol and cirrhosis related to NAFLD [43,45].

Semiannual surveillance for cirrhotic patients by ultrasound with or without α-fetoprotein (AFP) has been shown to improve clinical outcomes at a rational cost [133]. Several mainly retrospective observational studies have shown HCC surveillance to relate to early HCC detection and consequently to application of effective treatment and improvement in overall survival [134].

Still, whether HCC surveillance is effective remains a matter of controversy [135]. Data on utilization of current HCC screening recommendations in the overall cirrhotic population reveal rates less than 20%.

As demonstrated by several studies, the risk for HCC development is not uniform amongst all patients fitting within a particular patient subgroup. Therefore, the “one-size-fits-all” strategy for early HCC detection should not be the preferred screening approach for chronic liver disease patients. Risk stratification of patients within a disease subgroup has been already been reported based on clinical and laboratory features [12].

Moreover, HCC surveillance strategies of intermediate- and high-risk patients with cirrhosis are reported cost-effective and exceed those of current recommendations [136]. As illustrated in previous sections of this manuscript, refinement of screening strategies according to prevailing clinical and molecular risk factors is mandated in AILDs patients, as personalized HCC risk prediction will offer significant advantage in patients at high risk [137], although up to the present no validated biomarker has been defined as risk prediction factor for HCC in this population. This strategy has been effective in other cancer types, such as breast and colon cancer. Establishing surveillance methods and intervals according to individual risks will maximize benefits and reduce harm related to screening the whole population at risk for HCC [138].

## 4. How We Treat AILD Patients with HCC

Treatment of HCC is a task for a multidisciplinary team involving hepatologists, radiologists, surgeons, and oncologists [43,45]. Treatment options depend on the modified Barcelona Clinic Liver Cancer (BCLC) score that incorporates variables with prognostic value, including tumor status, liver function, and general health performance, and do not relate to the underlying etiology. [3,43]. Surgical options, such as liver resection and liver transplantation, are the pillars of treatment in cases with early-stage HCC, as they offer the best outcomes for these patients [3,43].

Extensive experimental and clinical research has led to significant breakthroughs related to pathogenesis, surveillance strategies, and treatment modalities in HCC [139].

Steroids in conjunction with azathioprine and other immunomodulatory drugs, such as mycophenolate myfetil (MMF), are the recommended first-line treatment for AIH patients [30,31,32,51,54,55,56]. Similar to other autoimmune diseases [140], long-term immunosuppressive treatment has been linked in limited reports to an increased overall risk for extrahepatic malignancies of various types in AIH patients compared to the general population [39,141]. In line with this, long-term immunosuppression has been long suggested as a risk factor for malignancies in organ transplant recipients. Derangement of cytokines and subsequent lymphocyte changes have been speculated to be the main cause of defective immune mediated tumor surveillance control. Apoptosis and growth of cancer cells as well as DNA damage and DNA repair mechanisms may also be impaired in an environment of chronic immunosuppression [142,143,144]. Moreover, cellular or molecular alterations induced or amplified by mechanisms underlying the autoimmune process per se may also enhance the risk of carcinogenesis.

Data from patients undergoing LT for HCC underscore the crucial role of immunosuppression type in patients’ outcomes by reducing the risk of HCC recurrence [145]. Moreover, in post-orthotopic LT, the incidence of de novo malignancies is 2–4 times higher than in the general population, with lymphoproliferative disorders and skin cancers being the most frequent ones [146].

Several studies have shown the pivotal role of immunosuppression type on the overall risk of malignancies post-LT. In fact, calcineurin inhibitors have been associated with increased HCC recurrence, while mammalian target of rapamycin (mTOR) inhibitors was shown to reduce HCC recurrence rates [145,146]. In fact, experimental data have shown mTOR to be overexpressed in a significant proportion of HCCs and mTOR inhibitors to exert tumor suppressive effects in HCC cells [147]. In addition, data related to outcome of patients post-LT have shown MMF to be associated with a reduction of HCC as well as less frequent development of de novo malignancies [148]. A recent study has shown MMF at a clinically relevant concentration to inhibit substantially the proliferation of human HCC lines and the development of colony forming units, while also providing evidence on reduced HCC recurrence and improved survival in a small number of patients receiving MMF [149]. Even though this study had several limitations and its results should be validated in larger randomized studies, it provides important information that could have great impact on the clinical management of patients post-LT. Whether these data on MMF can be extrapolated to the clinical management of AIH patients after they have developed HCC is something that needs to be explored further.

Our group has contributed toward establishing MMF as an alternative first-line therapeutic option for AIH patients, as this drug was linked to high off-treatment remission rates accompanied by improvement in liver histology [54,55,56,57]. Up to the present, we have no data to demonstrate whether MMF could have any beneficial effect in the overall incidence of HCC in AIH patients, as this would require a large patient database and a comparison set consisting of patients on azathioprine as primary treatment.

Based on this we cannot propose the modification of an immunosuppressive regimen in AIH patients after diagnosis of HCC as an adjunct treatment strategy. A future project would elaborate the possible beneficial effect of MMF on the natural history of HCC in AIH patients.

## 5. Conclusions and Future Perspectives

Overall, several studies have shown AILDs and especially AIH and PBC to confer increased risk to HCC, though lower compared to other hepatopathies. Several parameters have been recognized as risk factors for HCC in AIH and PBC patients, the most decisive being cirrhosis, older age, and male sex, as well as other co-existing factors, such as alcohol abuse. Recently, attention has been drawn to the crucial role of treatment response and disease remission as risk factors for HCC in AIH and PBC patients.

To date, scientific societies recommend surveillance mainly in cirrhotic patients, including patients with AIH and PBC. Considering that the risk for HCC varies between and within disease groups, surveillance should aim at screening intermediate- and high-risk patients as these strategies are reportedly cost-effective and exceed those of current recommendations.

Accordingly, future recommendations should incorporate clinical and laboratory features that have been acknowledged to confer increased risk for HCC in AIH and PBC patients.

Moreover, as data on the possible role of immunosuppression on the natural history of HCC pre- and post-diagnosis are lacking in AIH patients, future research should aim on further elaborating this issue.

## Figures and Tables

**Table 1 cancers-13-01023-t001:** Incidence of hepatocellular carcinoma (HCC) in patients with autoimmune hepatitis (AIH).

Study Publication Year	Country	Duration of Study	Patients *n*	Cirrhosis at AIH Diagnosis	HCC Cases	Total Follow Up Period	Follow Up to HCC Diagnosis	Incidence HCC Rate/1000 Person-Years	HCC during Follow Up
Miyake (2006) [68]	Japan	1988–2005	69	6%	3	96 (49–201) months	109 (108–169) months		4.3%
Floreani (2006) [69]	Italy	1981–2004	73	38.4%	1	71.1 ± 48.4 months			1.4%
Montano-Loza (2008) [70]	USA		227	23.3%	9	93 (12–372) months	122 (12–372) months	4	4%
Yeoman (2008) [62]	UK	1971–2007	243	48.6%	15	11 (1–36) years	102.5 (12–195) months	10.9	6.2%; 1.1%/year
Teufel (2009) [66]	Germany	1970–2008	278	11%	0	4.8 years (58 months)			
Wong (2011) [71]	USA	1999–2009	322		6				1.9%
Hino-Arinaga (2012) [72]	Japan	1978–2007	180	18.9%	6	80.2 ± 66.5 months	128.2 ± 89.9 months		3.3%
Yoshizawa (2012) [73]	Japan	1974–2010	203	12.8%	5	131 (13–432) months			2.5%
Migita (2012) [74]	Japan	1995–2008	193	10.9%	7				3.6%
Groenbeck (2014) [47]	Denmark	1994–2012	1721	28.3% (in 1318 patients)	13			0.8	
Van Gerven (2014) [50]	The Netherlands	1967–2011	730		8	8 (0–44) years	6 (9–29) years		1%
Danielson Borssen (2015) [75]	Sweden		634	28%	10	11.4 (0–40.5) years	14 (0–29) years		0.3/year in cirrhotic patients
Zachou (2016) [55]	Greece	2000–2014	159	31.6%	4	67 (3–168) months			2.5%
Puustinen (2019) [49]	Finland	1995–2015	887		11				1.24%
Choi (2019) [76]	S. Korea	2001–2017	291	34.7%	9				5.3% at 10 years

**Table 2 cancers-13-01023-t002:** Risk factors for hepatocellular carcinoma in patients with autoimmune hepatitis.

Study		Statistics
**Demographics**		
Older age	Macaron (2010) [78]	HR: 1.1, *p* < 0.001
	Dakhoul (2019) [67]	OR: 1.15, 95%CI: 1.001–1.314, *p* < 0.05)
	Choi (2019) [76]	*p* = 0.03
Male sex	Montano-Loza (2008) [70]	HR: 7.0, 95%CI: 1.87–26.1, *p* = 0.004
	Migita (2012) [74]	*p* = 0.033
	Grønbæk (2014) [47]	Males vs. females: 2.1 (0.8–4.7) vs. 0.3 (0.0–1.0) HCC incidence rate per 1000 person years (95%CI)
**Clinical features**		
Cirrhosis	Montano-Loza (2008) [70]	Cirrhosis for ≥10 yr HR: 8.4; 95%CI: 1.69–41.9, *p* = 0.009
	Yeoman (2008) [62]	Cirrhosis at presentation of AIH, *p* < 0.05
	Hino-Arinaga (2012) [72]	OR: 4.08; 95%CI: 1.54–18.32, *p* = 0.001
	Migita (2012) [74]	HR: 11.47; 95%CI: 2.13–64.6, *p* = 0.005
	Choi (2019) [76]	*p* < 0.001
	Jensen (2020) [65]	HR: 7.6; 95%CI: 3.3–17.8
Any sign of portal hypertension (ascites, thrombocy-topenia, or esophageal varices)	Montano-Loza (2008) [70]	HR: 19.1; 95%CI: 3.91–93.3, *p* = 0.0003
	Yeoman (2008) [62]	Variceal bleeding at presentation, *p* = 0.003
	Vaz (2020) [79]	Portal hypertension at diagnosis; HR: 4.88; 95%CI: 1.49–15.92, *p* = 0.009
Thrombocytopenia	Montano-Loza (2008) [70]	HR: 7.3; 95%CI: 1.89–28.3, *p* = 0.004
**Data related to biochemical activity and treatment response**		
Abnormal ALT at last follow up Disease remission ≥2 relapses	Hino-Arinaga (2012) [72] Vaz (2020) [79] Yoshizawa (2012) [73]	OR: 3.66; 97%CI: 1.38–16.42, *p* = 0.02 Lower risk of HCC occurrence HR: 0.128; 95%CI: 0.043–0.38, *p* < 0.001 HR: 9.1; 95%CI: 1.8–45.5, *p* = 0.007
**Co-existing factors**		
Alcohol consumption	Macaron (2010) [78] Rigopoulou (2021) [29]	HR: 5.9, *p* < 0.001 No statistically significant difference

OR: odds ratio; HR: hazard ratio.

**Table 3 cancers-13-01023-t003:** Risk factors for hepatocellular carcinoma in patients with primary biliary cholangitis.

Study		Statistics
**Demographics**		
Older age	Shibuya (2002) [102]	*p* = 0.008
	Suzuki (2007) [105]	OR: 1.7; 95%CI: 1.1–2.5, *p* < 0.01
	Deutsch (2008) [106]	HR: 1.16; 95%CI: 1.02–1.32, *p* = 0.027
	Rong (2015) [107]	Age > 54 years OR: 5.5; 95%CI: 3.0–10.1, *p* = 0.001
Male sex	Shibuya (2002) [108]	*p* = 0.005
	Suzuki (2007) [109]	OR: 9.7; 95%CI: 1.4–68.3, *p* = 0.02
	Harada (2013) [101]	OR: 3.09, *p* = 0.0012
	Rong (2015) [107]	OR: 2.2; 95%CI: 1.2–4.0, *p* = 0.001
**Clinical features**		
Histological stage	Shibuya (2002) [108]	Higher incidence in advanced (stage III/IV) vs. early stage (I/II) *, *p* = 0.021
	Cavazza (2009) [110]	Higher incidence in advanced stage OR: 5.80; 95%CI: 2.34–14.38 * ^#^, *p* < 0.001
	Deutsch (2008) [106]	Baseline stage IV, HR: 31.50; 95%CI: 1.59–625.30 ^#^, *p* = 0.024
	Harada (2013) [101]	Increase in incidence with histological stage in females *, *p* < 0.001
Any sign of portal hypertension (ascites, thrombocypenia, or esophageal varices)	Suzuki (2007) [109]	OR: 22.9; 95%CI: 3.4–155.3, *p* < 0.01
Thrombocytopenia	Trivedi (2015) [103]	HR: 1.41; 95%CI: 1.25–1.58, *p* < 0.0001
**Data related to biochemical activity and treatment response**		
Non-response to UDCA	Kuipers (2010) [111]	*p* < 0.001
	Trivedi (2015) [103]	Biochemical non-response (Paris-I criteria) HR: 3.44; 95%CI: 1.65–7.14, *p* < 0.0001
**Co-existing factors**		
Past HBV infection	Rong (2015) [102]	OR: 6.6; 95%CI: 3.7–11.9, *p* = 0.001
Diabetes mellitus	Rong (2015) [107]	OR: 3.1; 95%CI: 1.6–6.2, *p* = 0.002
BMI ≥ 25 (kg/m^2^)	Zhang (2015) [112]	OR: 1.116; 85%CI: 1.002–1.244, *p* < 0.05
Alcohol consumption	Zhang (2015) [112]	OR: 10.294; 95%CI: 1.108–95.680, *p* = 0.04

OR: odds ratio; HR: hazard ratio. * Histological stage was classified according to Scheuer’s criteria; ^#^ Histological stage was classified according to Ludwig criteria.

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
