# Peer review of "Current Trends and Characteristics of Hepatocellular Carcinoma in Patients with Autoimmune Liver Diseases"

_cancers, 2021, doi:10.3390/cancers13051023_

Round 1

Reviewer 1 Report

In the current review, Rigopoulou et al. provided an extensive description of the risk factors for hepatocellular carcinoma in the patients with autoimmune liver diseases. The authors summarize the most resent and relevant literature. The review is well-organized and will provide an outstanding resource for researchers studying hepatocellular carcinoma.

I only have a few suggestions that if properly addressed by the authors may increase the significance and value of this review.

  1. Additional table summarizing the risk factors for hepatocellular carcinoma or intrahepatic cholangiocarcinoma in patients with primary sclerosing cholangitis as with AIH and PBC.

  1. Section 3 is a mix of content that confuses the reader. It is recommended to re-organized section 2 and 3. Variant syndromes should be included in section 2. In that case, section 3 can be about screening and treatment strategies for hepatocellular carcinoma.

  1. Most of section 3 only mentions general screening strategies for hepatocellular carcinoma. It is advisable to mention a little more about strategies specific to autoimmune liver disease. The same is recommended for treatment strategies in section 3.2.

Author Response

  1. We would like to thank the reviewer for this comment. However, we think that the inclusion of a table referring to risk factors for HCC or intrahepatic cholangiocarcinoma in PSC patients, as we have done for AIH and PBC patients would be of low priority and would not add to the overall scientific value of the manuscript, as the review refers to HCC and not intrahepatic cholangiocarcinoma. Moreover, as we have already reported in the relevant section of the manuscript, there are very few studies reporting on the incidence of HCC in PSC patients, while no risk factors have been acknowledged so far.
  2. We have followed the advice of the reviewer and we have now reorganized sections 2 and 3.
  3. We appreciate the reviewer’s comment. Indeed, section 3 refers to general screening strategies for HCC, as there are no specific treatment strategies for patients with AILDs up to the present. This is now added in the respective section of the manuscript (section 3, 1st par.). For this reason and based also on data provided in individual sections, we believe that screening strategies should be adapted according to existing risk factors and propose that screening should be prioritized for groups of patients considered to be at highest risk. We think that this is clearly stated in the last paragraph of section 3. “As illustrated in previous sections of this manuscript, refinement of screening strategies according to prevailing clinical and molecular risk factors is mandated in AILDs patients, as personalized HCC risk prediction will offer significant advantage in patients at high risk [137]. This strategy has been effective in other cancer types, such as breast and colon cancer. Establishing surveillance methods and intervals according to individual risks will maximize benefits and reduce harm related screening for the whole population at risk for HCC [138]”.

    The same applies to treatment strategies for HCC (now section 4), as there are no treatment recommendations specific to AILDs. We stress this in the first paragraph of section 4 (line 4).

    In addition, we refer to the pοtential role of immunosuppression and type of immunosuppression as adjunct treatment after HCC diagnosis, based on data on HCC recurrence post liver transplantation (p.12; 3rd par.). 

Reviewer 2 Report

Reviewer’s comments

This review article primarily focused on carcinogenesis in autoimmune liver disease. The article seems to be quite impressive. It provides us many current trends in patients with autoimmune liver disease who developed HCC. I really appreciate your efforts. However, several revisions are required for publication. Please refer to the comments shown below.

Major

#1. Is the incidence of HCC development in patients with autoimmune liver disease currently increased, decreased, or unchanged? The authors should describe the incidence in “Introduction”.

#2 One of the risk factors for HCV-related HCC is also cirrhosis. However, the incidence of HCC in HCV-related liver cirrhosis is much higher than that in AIH-related cirrhosis. It is one of the most important points in this review article. The authors should describe or speculate the reason.

#3. Clinical characteristics of type 2 AIH are distinct from those of type 1 AIH. Patients with type 2 AIH more frequently develop relapse than those with type 1 AIH. Therefore, I guess that the incidence of HCC in patients with type 2 AIH may be higher than that in patients with type 1 AIH. The authors should compare the incidence between the groups.

#4. It has been well documented that chronic HCV infection evokes varieties of autoimmune phenomena, including autoantibodies’ production and concurrent extrahepatic autoimmune disease. However, this review article does not describe HCV-induced autoimmune hepatitis at all. The incidence of HCC in patients with HCV-induced AIH should be also revealed.

#5. It is of interest that the difference in the incidence of HCC was not found between concurrent AIH/NASH patients and AIH patients. The authors should describe the clinical characteristics of AIH patients concomitant with NASH who developed HCC. (For example, male dominant ?, insulin resistance ?)

#6. Obesity and concurrent diabetes seemed to be risk factors for HCC development in patients with PBC. However, these factors were not identified in patients with AIH as the risk factors. The authors should explain the difference between the diseases.

#7. The statement, “Long-term immunosuppressive treatment has been linked to an increase overall risk for extrahepatic malignancies in AIH patients” is quite interesting to me. The authors should describe or speculate the reason.

Minor

#1. “Total follow time” should be corrected to “total follow-up period” in Table 1.

#2. The statistics in Migita’s report, “p=0.033” should be moved to lower row in Table 2.

#3. “HR” and “OR” should be spelled out, respectively in Table 2.

#4.Page 6, line 24, “Defective Immune mediated・・“should be corrected to ”Defective immune-mediated・・・“.

#5. The authors should mention which classification was used as a histological stage. Scheuer’s staging system or Ludwig’s histological classification?

#6. “BMI (kg/m2)” should be corrected to “BMI (kg/m2)” in Table 3.

#7.Page 10, line 49, “a-fetoprotein” should be corrected to “α-fetoprotein”.

Author Response

1. We would like to thank the reviewer for his comment. However, data on the evolving trend related to incidence of HCC in autoimmune liver diseases as a whole, are inconsistent. This relates to the fact that autoimmune liver diseases encompass a wide spectrum of liver diseases with different pathophysiology, clinical presentation, treatment options and prognosis and also to the heterogeneity of published studies. Besides, studies elaborating the incidence of HCC in AILDs vary considerably to allow clear assumptions on the epidemiological trends of HCC in these group of patients. Therefore, we would like this topic to stay as it is presented in the first version of the manuscript (incidence rates of HCC in each respective AILD).

  1. We absolutely agree with you that the incidence of HCC in HCV-related liver cirrhosis is much higher than that in AIH-related cirrhosis. Several studies have elucidated up to the present direct and indirect interactions of HCV with cellular components in cirrhotic liver that promote hepatocarcinogenesis. Furthermore, data in experimental animal models have shown HCV proteins, including HCV core, NS3, NS4B and NS5B to have oncogenic properties and eventually transform specific cells types. These issues have now been added in the revised version along with 3 new references (please see p. 2, sixth par. and new ref. 16-18).
  2. We appreciate the reviewer for the comment. Even though differences between type 1 and type 2 AIH patients in terms of demographics, clinical presentation and outcome as long as treatment response exist, there is no data up to the present to support differences in HCC incidence between the two groups. This is now included in the section “2.1 Autoimmune hepatitis”, page 6; last par.
  3. Again we would like to thank the reviewer for this comment. However, it should be stated that HCV-induced AIH has been foremost reported in IFN-a-treated patients in the past, when IFN-a was the backbone of HCV treatment. Up to the time of this writing, data on the potential role of direct acting antivirals (DAAs) in inducing autoimmune phenomena are scarce to draw firm conclusions. While several studies exist reporting on the role of autoantibodies (organ-specific and non-organ-specific) on histology and treatment response, to our knowledge there are no data in the literature on the natural history of HCV- induced AIH, including incidence of HCC. This has now been added in a sentence (p. 6, last sentence).
  4. We would like to thank the reviewer for this valid comment. Unfortunately, there are no data from the available literature that report on specific characteristics of patients with AIH and NAFLD who developed HCC. The largest study published as full paper up to the present by Takahashi and colleagues (now reference 84), indicated that patients with AIH/NAFLD were in general less frequently female and older than AIH patients. Though no information on characteristics of those patients with AIH/NAFLD that developed HCC is reported in any of the three studies presented in the manuscript (ref. 84-86). This is now clearly stated in page 7 (end of 1st paragraph)
  5. We appreciate the reviewer's remark. Obesity as well as alcohol have been reported as risk factors for the development of HCC also in other chronic hepatopathies, including HBV and HCV infections. Based on these facts, differences in risk factors for HCC between AIH and PBC patients may probably indicate differences in populations studied and subsequently in differences in individual characteristics of patients included in the studies rather than discrepancies related to specific hepatopathies per se (please see p.9).
  6. According to your recommendation a more detail reference on the mechanism underlying carcinogenesis has been now added in the relevant section together with the respective references (Page  12).

Minor revisions

  1. ‘Total follow time” has been replaced by “total follow up period” in Table 1.
  2. We thank the reviewer for this comment. Several adjustments have been now made in Tables 2 and 3.
  1. “HR” and “OR” have been now spelled out respectively in Table 2 and Table 3.
  1. We have now replaced “Defective Immune mediated” with “Defective immune-mediated” in page 6.
  1. We have now included in Table 3 the classification system used to classify the histology stage in PBC patients in every study reported in this Table.
  1. “BMI (kg/m2)” is now corrected to ‘BMI (kg/m2)”

7. “a-fetoprotein” is now corrected to “α-fetoprotein”

Reviewer 3 Report

I read the review from Dr. Rigopoulou and Dr. Dalekos. The authors submitted a comprehensive study entitled "Current Trends and Characteristics of Hepatocellular Carcinoma in Patients with Autoimmune Liver Diseases". The manuscript needs to be streamlined. The argument is interesting and certainly needs precise development in all fields, but the work results are too long, in my opinion. 

Author Response

We would like to thank you for your kind words regarding our review. However, we should state that the paper was written according to the instructions of this journal, which needs at least 4.000 words and 2 tables excluding references and tables. So, we believe that our manuscript is not too long for this specific journal taking also into consideration that several additions have been suggested -almost obligatory- by the other reviewers. However, in order to satisfy at least partly the reviewer, we have now added a box summary at the end of the article in an attempt to streamline the manuscript and stress the basic points of the review.   

Reviewer 4 Report

This review manuscript investigated some risk factors between AIH and HCC, including older age, male sex, cirrhosis and alcohol consumption. Biochemical activity and treatment response are prognostic factors. Surveillance can detect HCC at earlier stage. Refinement of screening strategies is mandated in AILDs patients. Personalized risk prediction will offer significant advantage in high- and/or medium risk. Modification of immunosuppression could also benefit AIH patients after HCC diagnosis.

Major comments,

The manuscript is well prepared and written and useful for clinical reference.

Minor comments,

  1. In table 3, was HCV infection a risk of HCC development in primary sclerosing cholangitis? Because HCV is usually associated with other systemic disease.
  2. The risk factors for HCC in AIH and PBC patients, the most decisive being cirrhosis, older age and male sex as well as other co-existing factors, such as alcohol abuse. Is there any biomarker being a risk factor?

Author Response

Minor comments

  1. HCV infection hasn’t been identified as a risk factor for HCC in PBC patients.

2. We would like to thank the reviewer for this comment. To our knowledge there is not any biomarker up to the present to have been identified as a risk factor for any of the AILDs. This statement has now been added in the revised text (p.11).

Round 2

Reviewer 1 Report

The manuscript entitled "Current trends and characteristics of hepatocellular carcinoma in patients with autoimmune liver diseases" by Eirini Rigopoulou et al. has been significantly revised and is considered suitable for publication.

Reviewer 3 Report

The authors have reply to all reviewer's comments. The manuscript seems acceptable for pubblication now.